# A comparison of the cost-effectiveness of the current standard of care and of the haematology analyser XN31-microscopy combination for diagnosing malaria in a nonendemic country

Stephane Picot[1,2,+], Anne-Lise Bienvenu[1,3]

[1]University Lyon, Malaria Research Unit, Institute of Molecular and Supramolecular Chemistry and Biochemistry, Lyon, France
[2]Hospices Civils de Lyon, Hôpital de la Croix-Rousse, Institute of Infectious Agents, Department of Parasitology and Medical Mycology, Lyon, France
[3]Hospices Civils de Lyon, Groupement Hospitalier Nord, Service Pharmacie, Lyon, France

**BACKGROUND** The biological diagnosis of imported malaria cases is a challenge that requires efficient methods, trained staff and high-quality proficiency. Microscopy, rapid diagnosis tests and molecular tests are widely available and provide excellent results. New methods such using haematology analysers have been recently developed.

**OBJECTIVES** In the context of limited resources, a complete cost-effectiveness analysis of the different scenarios should aid in the decision-making process for the most appropriate methods.

**METHODS** The full cost-effectiveness of each malaria diagnosis scenario relative to the clinical benefits of the outcome was measured. The study population was a cohort of patients who were receiving health care at Lyon University Hospital for suspected malaria during 2023. Four scenarios were tested: microscopy, rapid diagnosis test + microscopy combination, loop-mediated isothermal amplification (LAMP) + microscopy combination, all identified as the control tests, and haematology analyser XN-31 + microscopy combination, identified as the intervention. The direct costs were calculated based on prices paid in France for material and consumables needed to perform malaria diagnosis for one sample among 1000 tests per year. The indirect cost of technical training, supervision and quality proficiency was calculated based on the hourly salary of the laboratory technician, junior and senior doctors according to the time needed for each scenario.

**FINDINGS** This approach provides a global approach for determining the cost-effectiveness of the most frequent scenarios for diagnosing malaria. The diagnosis accuracy and the short time-to-result of the haematology analyser combined with microscopy were the key points of the cost-effectiveness result.

**MAIN CONCLUSION** The generalisability of our findings is restricted by the specificity of the costs observed in France and the limited panel of methods tested. However, this may promote similar studies from other countries to document the cost-effectiveness of the different scenarios used for malaria diagnosis.

Key words: malaria diagnosis - cost-effectiveness - microscopy - rapid diagnostic test - loop mediated isothermal amplification - XN-31 - haematology analyser

In malaria-endemic areas, diagnosis relies on simple and effective methods, including rapid diagnosis tests (RDTs) and the microscopic examination of Giemsa-stained thick or thin blood smears. However, in non-endemic areas, most laboratories receive a low number of suspected malaria cases, and the positivity rate among those is low.[1,2] Consequently, the gradual disappearance of experienced microscopists impairs malaria diagnosis based on microscopy, especially for non-*falciparum* species. This leads to variation in the limit of detection (LoD) from 5 to 100 parasites/μL.[3-7] Thus, achieving and maintaining proficiency in the microscopic examination of stained blood smears is time-consuming.[8]

Similarly, RDTs have several limitations, including reduced sensitivity for non-*falciparum* malaria species. Furthermore, the emergence of parasite clones lacking HRP2/3 proteins is associated with a risk of false-negative results with RDT using this marker.[9]

The cost of molecular methods has decreased significantly, making them highly efficient at detecting submicroscopic parasitaemia in asymptomatic patients. Many different tools based of DNA amplification have been developed recently.[10] One such rapid molecular method widely used by diagnostic laboratories in non-endemic areas is the loop-mediated isothermal amplification (LAMP).[11-14] Although the polymerase chain reaction

Financial Support: This study was supported by a grant from Sysmex Corp, Japan. The Malaria Research Unit is an associated laboratory of the University Lyon 1 (UCBL1) and the National Centre for Scientific Research (CNRS) and receives financial support from UCBL1 and CNRS. SP received grant support from The European & Developing Countries Clinical Trials Partnership (EDCTP2, WANECAM2 RIA2017T-2018), a two grants from the French National Research Agency (ANR-22-CE19-0026 MALA-EYES and ANR-21-CE09-0011-03).
+ Corresponding author: stephane.picot@univ-lyon1.fr | ⓘ https://orcid.org/0000-0002-5735-6759

**How to cite**: Picot S, Bienvenu A-L. A comparison of the cost-effectiveness of the current standard of care and of the haematology analyser XN31-microscopy combination for diagnosing malaria in a nonendemic country. Mem Inst Oswaldo Cruz. 2025; 120: e250078.
**Handling editor**: Cláudio Tadeu Daniel-Ribeiro | ⓘ https://orcid.org/0000-0001-9075-1470

(PCR) is the gold standard for detecting and identifying *Plasmodium*, this method cannot be used for emergency diagnosis, when the time taken to obtain a result may affect the prognosis of the disease. In France, it is mandatory to provide a result for malaria diagnosis within 2 h of receiving the blood sample in the laboratory, 24 h a day, seven days a week. PCR does not meet these requirements, and was not included in this study. However, all these methods require trained personnel and equipment to minimise the risk of cross-contamination associated with very high sensitivity.

The application of artificial intelligence (AI) to malaria diagnosis is promising and fast-growing area of activity. However complex issues have not yet been fully addressed, and it is likely to take longer than expected to develop efficient methods for routine diagnosis.[15] Over the past few years, haematology analysers have been developed and for use in both malaria-endemic and malaria-non-endemic areas. The most advanced technology in this field is based on the XN-31 analyser from the Japanese Company Sysmex, which is commercially available and has been successfully tested in various countries, including non-endemic regions such as France and Japan.[16-19] Indeed, we previously conducted a prospective study to evaluate the diagnostic accuracy of the XN-31 for diagnosing and monitoring imported malaria cases compared to reference malaria tests including RDT, LAMP, and microscopy.[20,21] We demonstrated that the XN-31 analyser meets the technical requirements to be considered in malaria diagnostic strategies in well-equipped laboratories. However, although XN-31 is an attractive method, it cannot be used as a standalone method. Full identification of non-*falciparum* species (*P. vivax*, *P. ovale*, and *P. malariae*)[22] is recommended to address the need to prevent the recurrence of *P. vivax* via specific treatment with primaquine after the resolution of active malaria cases.[23,24]

Thus, the global strategy for malaria diagnosis is complex, and many factors must be considered, including the accuracy of diagnosis, the proficiency of microscopist, the incidence of malaria cases and the final cost of the entire diagnosis process. Different workflows can be implemented in laboratories in areas where malaria is non-endemic based on the combinations of two or more of the most widely available methods. Microscopy and XN-31 are the only methods providing species identification or partial identification (*falciparum*/non-*falciparum*) respectively, and parasitaemia from the same test. In the context of limited resources for medical care, including in high-income countries, the cost-effectiveness of a diagnostic workflow is a key consideration for decision-makers and final buyers.[25] This study primarily aims to document the cost-effectiveness of the most current standard of care for malaria diagnosis according to a performance-based decision tree model. The secondary objective is to compare the status quo with scenario involving XN-31 combined with microscopy.

### SUBJECTS AND METHODS

*Study site and population* - At the national level in France, approximately 4,000 to 6,000 malaria cases are imported each year. Lyon University Hospital (Hospices Civils de Lyon), with 13 different hospitals in Lyon city and its suburbs and 24,000 employees, is the second most important medical centre in France. The study population was based on the cohort of patients receiving healthcare at Lyon University Hospital for fever and suspected malaria in 2023. All the departments receiving malaria cases were included in the cohort. The following were documented for the included patients: age, Plasmodium species, hospitalisation levels [intensive care unit (ICU), non-ICU], and positive or negative outcomes. This research involved anonymised data sets in which personal identifiers were permanently and completely removed from the data, meaning that the data can no longer be associated with an individual in any way. Electronic records were stored and processed in compliance with the conventions of the French National Committee for Data Protection and Freedom of Information.

*Ethical clearance* - This research involved anonymised data sets in which personal identifiers were permanently and completely removed from the data, meaning that the data can no longer be associated with an individual in any manner. Electronic records were stored and processed in compliance with the conventions of the French National Committee for Data Protection and Freedom of Information. Ethical approval for previous comparison of malaria diagnosis methods was obtained from the Scientific and Ethics Committee of Lyon University Hospital (Avis CSE n°20_151). Cost-effectiveness research was not related to patient data and thus did not require ethical approval.

*Biological tests and strategies for malaria diagnosis* - Most of the laboratories in non-endemic countries use a combination of microscopy, RDTs and LAMP or PCR for diagnosis confirmation or species identification, according to French National guidelines from the High Authority of Health. Similar guidelines are produced in other non-endemic European countries, such as the UK.[26] Microscopy is the reference for all scenarios (comparators and interventions) for the post-admission follow-up. A comparator scenario using both RDT + LAMP for screening followed by microscopy for species identification and parasitaemia was not tested as a standard-of-care because of its high cumulative costs compared to other scenarios, while its effectiveness is optimal.

*Health economic analysis plan* - The study was conducted in agreement with the Consolidated Health Economic Evaluation Reporting Standards (CHEERS) 2022 checklist and recommendations (2022 statement: updated reporting guidance for health economic evaluations) and following the recommendations from the B&M Gates Foundation Report.[27,28]

The full cost-effectiveness of each malaria diagnosis method relative to the clinical benefits of the outcome was measured in monetary and non-monetary values. We developed a decision-analytical model to estimate the incremental costs and outcomes of the introduction of the XN-31 analytical process combined with microscopy in a real hospital setting ("intervention") against a series of comparators of standard diagnosis methods using either

microscopy, RDT, or molecular tests of a combination of these ("controls"). We modelled cost-effectiveness separately using the diagnostic parameters from each of these methods. A decision tree was built to compare the tested scenarios according to the performance of each method (true positives/false negatives) and their impact on patient outcomes using Decision Tree software 6.1.10 (from Spicelogic, Inc., Thornhill, Canada).

Payer (National Health Insurer) and Societal (Quality Adjusted Life Years: QALY) perspectives were used in this study. The time horizon was limited to the most frequent evolution of a malaria case in a non-endemic area, without recrudescence, reinfection, or long-term complications. A standardised discount rate was not applied since the time horizon was less than one year. The currency used was Euro (€).

*Intervention and model structure* - A general practitioner usually prescribes a malaria test to a febrile patient with a history of travel to an endemic area. In the case of a positive test, the patient was referred to the hospital for confirmation and treatment. As many private clinics do not treat emergency cases of travellers presenting with a fever, this option was excluded from the analysis. Most of the referred patients are admitted to the emergency department for several hours, depending on the severity of the disease, before being transferred to other departments. Patients with severe malaria, are transferred to the ICU until clinical improvement. This study focused on the economic analysis of public hospital activities. The final outcomes of febrile patients suspected of having malaria from initial presentation to the hospital setting, through diagnosis, treatment, disease progression and further care, were considered.

For the measurement of outcomes, we estimated the success of initial treatment in curing malaria in our specific setting using negative parasitaemia on day 3 ("initial treatment effectiveness") and day 28 ("final treatment effectiveness") according to World Health Organisation (WHO) recommendations.[29] In the case of *P. falciparum* malaria, we assumed that the illness might progress to a severe case based on real malaria cases included in the study. In the case of non-*falciparum* malaria, we assumed that the illness resolved without further care, and we considered a risk of severe cases lower than 1%.

Since all *Plasmodium* species are treated in the same way for non-severe malaria and that severe malaria cases are almost exclusively caused by *P. falciparum*, the decision tree analysis included the exact identification of non-*falciparum* malaria cases requiring microscopic examination by a senior doctor. The final step in the biological diagnosis process was identifying the presence of *P. falciparum* or non-*falciparum* as well as determining the level of parasitaemia, which is required at admission and for treatment follow-up. The impact of *P. falciparum*/non-*falciparum* and severe/non-severe heterogeneity was assessed.

The analysis also took into account the potential discounts that manufacturers could easily offer to specialised laboratories, as well as heterogeneity in sale prices. This could significantly affect the global cost of the LAMP test. We used catalogue prices for the main analysis, and tested this heterogeneity secondarily.

The criteria for the analysis were as follows: minimum possible cost: 4 €; maximum possible cost: 5000 €; and maximum willingness to pay for each additional unit of effectiveness: 50 €. The effectiveness criteria were the maximum QALY, minimum cost-effectiveness ratio, and net monetary benefit.

*Data sources* - The numbers of uncomplicated and severe cases per year at the study site were collected during the year 2023. The positivity rates of methods used in our setting and the probabilities of true positives/false negatives were extracted from our previously published studies conducted in the same place with similar methods.[13,14,20] We used references from the WHO for microscopy performance and from FIND/WHO reports for RDT accuracy.[30] The performances (%) of analytical tests were: Microscopy: sensitivity 93.9 (83.1-98.7) and specificity 100 (98.9-100);[11] RDT: sensitivity 95.0 (93.5-96.2) and specificity 95.2 (93.4-99.4);[29] LAMP: sensitivity 90.24 (76.9-97.3) and specificity 96.65 (89.2-98.8);[31] XN-31: sensitivity 100 (97.13-100) and specificity 98.39 (95.56-100).[20]

The valuations of resources and costs were extracted from the French Technical Agency for Hospitalisation Information (ATIH). The ATIH provides the real cost of the main diseases declared by French hospitals each year. Malaria patients treated at the hospital have different costs depending on the severity and length of hospital stay. Cases are classified into four degrees with specific medical and hotel costs as follows: very short hospitalisation time (outpatients): 1041 € (882-1201); Level 1: 2495 € (2282 - 2708); Level 2: 3901 € (3577 - 4224); Level 3: 7129 € (5849 - 8409).

To estimate the cost of a false negative or false positive result as accurately as possible, the time taken to make the correct diagnosis and start treatment was considered. The cost of this delay is difficult to evaluate precisely because it depends on the cost of undiagnosed (false negative) or over diagnosed (false positive) diseases. In the case of a false negative result, a malaria patient's condition may deteriorate rapidly, requiring an increase in the level of care, from level 1 to level 2. Therefore, we considered that the impact of the delay is represented by the difference in total costs paid by the French government for a non-severe malaria patient hospitalised in a level 2 ward (€3,901, range €3,577-€4,224) compared to a level 1 ward (€2,495, range €2,282-€2,708). We therefore applied an additional cost of €1,406 to each false negative result. A false positive result may lead to the inappropriate treatment of patients with antimalarial drugs, as well as a delay in diagnosing and treating non-malaria cases caused by fever. Due to the wide range of underlying causes of fever, we attributed the same additional cost (€1,406) to false-positive results. The diagnosis and supporting intervention costs were based on real costs: 1) the costs of the intervention and comparators, including investments and consumables (actual costs in France based on real quotes); 2) the cost of initial training and quality assurance proficiency

testing, according to French requirements. Data on the time spent on training and testing was collected through interviews with malaria field workers.

*Intervention and control tests direct costs* - The costs of the intervention and control tests were calculated based on the prices paid (taxes excluded) by the Lyon University Hospital in 2023 for all consumables, quality controls, and one dedicated machine with a specific depreciation rate and maintenance needed to perform malaria diagnosis for one sample among 1000 tests per year (Table I).

The cost of the intervention was calculated based upon the price of all reagents needed to operate the equipment, including the cost of daily quality control. These costs were extracted from quotes paid in France in 2023. The XN-31 sale price was 30,800.00€ (tax excluded), and the annual maintenance price was 6,000.00€ (tax excluded). Using a depreciation over five years, the total cost of equipment was 12.16 € per test for 1000 tests per year. The test requires 6 different reagents to be used, including internal controls. The limit of validity of each reagent was defined as less than 90 days. It was speculated that 250 tests can be performed per trimester. The total cost of reagents to perform one test was 12.2 €. Globally, the full direct cost of performing one malaria test with the XN-31 automate was 12.16 ± 12.2 = 24.36 €.

Microscopy was performed after the completion of thin and thick blood smears on a microscope slide. The cost of dropping and smearing blood on a glass microscope slide with a polished edge was based on the use of three different slides (for smearing and microscope observation). The sale price of a standard box of 50 slides is 5.0 € (3.45 - 6.20€), which is 0.30 € for one test. The staining costs were the price of a bottle of stain divided by the volume used to stain three slides of the same sample. Thin blood smears were stained with either RAL Diff-Quick [120.48€ HT (3 x 500 mL) = 0.08 € per test] or Giemsa R [(23.5€ HT/500 mL) = 0.047 € per test]. Giemsa is the most popular stain used for this study. The cost of the microscope was based on the last quote obtained in France for a microscope used on a daily basis for diagnosis (4,890.00 € and 250.00 € for yearly maintenance, tax excluded). High heterogeneity can be suspected for this equipment sale price depending on the optical performance, and optical microscope prices can be highly variable. The cost of six external quality controls per year was added. The total direct cost for microscopy was 2.13 € for one test on the basis of 1000 tests per year.

The LAMP test used the Alethia technology for Malaria (Meridian Bioscience, distributed by Launch Diagnostics in France). The complete direct cost of 45.0€ per test included the equipment (Alethia, Alethia printer, verification standards and maintenance) and the cost of reagents (SMP PREP KIT 25 tests and Alethia Malaria 18 controls). Based on our experience with potential cross-contamination between samples, we considered that one control should be tested every 10 samples, meaning that 100 controls are needed per year for 1000 samples tested. During this study, we used the public sale prices for equipment, reagents and controls. We subsequently rerun the analysis model with the same decision tree using the reduced prices obtained from the provider for the equipment, controls and reagents for the heterogeneity analysis.

The RDT was PALUTOP® +4 OPTIMA from Biosinex (Illkirch Graffenstaden, France). According to the FIND/WHO, this test is one of the top RDTs available on the market. The price for one test is 3.88 € without additional cost for sample preparation. The quality control was based on four lyophilised blood samples provided at an annual cost of 184.00 €, representing an additional cost of 0.18 € per test. The total direct cost for RDT is 4.06 € per test. A main comparator is the CareStart Malaria HRP2/pLDH (Pf/pan) combo, which is also available on demand in our setting, with a very similar cost of 3.77 € per test.

*Indirect costs: workload and work costs* - The costs of technical training, supervision and quality proficiency were calculated based on the hourly salaries of laborato-

TABLE I

Direct costs of malaria diagnostic methods in France (2023). Equipment depreciation was calculated over five years. The depreciation, maintenance and quality control costs were calculated using real quotes for 1,000 tests per year and were calculated in the same way for all methods

| Direct costs | Microscopy | | RDT | | LAMP | | XN-31 | |
|---|---|---|---|---|---|---|---|---|
| Equipment | 4,890.00 | | 0.00 | | 22,000.00 | | 30,800.00 | |
| | /year | /test | /year | /test | /year | /test | /year | /test |
| Depreciation/year over five years | 978.0 | | 0.00 | | 4,400.0 | | 6,160.0 | |
| Depreciation/test | | 0.98 | | 0.00 | | 4.40 | | 6 .16 |
| Maintenance/year | 250.0 | | 0.0 | | 1,050.0 | | 6,000.0 | |
| Maintenance/test | | 0.25 | | 0.00 | | 1.05 | | 6.00 |
| Reagents & QC tests | | 0.90 | | 4.06 | | 39.55 | | 12.20 |
| TOTAL direct cost/test | | 2.13 | | 4.06 | | 45.00 | | 24.36 |

RDT: rapid diagnosis test; LAMP: loop-mediated isothermal amplification.

ry technicians, junior and senior doctors according to the time needed for each scenario based on local experience and the National Reference for cost of care (Référentiel National des coûts de prise en charge). The 2023 salary costs of professionals involved in biological tests in France were obtained from national references and documented sources from the French Ministry of Public Service.

The workload was defined as the number of hours needed for training for each specific method according to the level of expertise required for technicians, junior doctors and senior doctors (Table II).

Operational execution costs were the addition of the work cost of the laboratory technician + junior + senior doctors for each test, weighted by the time needed for each operator. It was estimated from local experience

that the cumulative work time of a laboratory technician, a junior and a senior doctors, needed to train and qualify the three workers was 26 h (6 + 8 + 12) for microscopy (species identification and parasitaemia), 7 h (2 + 2 + 3) for RDT, 10 h (3 + 3+ 4) for LAMP and 12 h (4 + 4 + 4) for the XN-31. The quality proficiency costs were estimated on a quarterly basis and were counted once for every 1000 tests.

*Total cost (direct + indirect) of biological diagnosis (controls and intervention)* - The costs of the controls and interventions were the addition of the direct and indirect costs for each test (Table III). Direct costs were the sum of the single test cost plus the cost of equipment when needed, with a depreciation of 20%/year and a mainte-

TABLE II

Indirect costs of biological malaria diagnostic methods in France (2023). Hourly work costs (including taxes) were obtained from public health authorities for laboratory technicians, and doctors (junior and senior). Training, quality proficiency and execution times were obtained through personal experience and interviews with professionals

| Work cost/hour (€) | Lab. technician | Junior Dr | Senior Dr | |
|---|---|---|---|---|
| | 18.86 | 14.02 | 49.26 | |
| Training & trimestral quality proficiency test | | | | Cost/year (1000 tests) (€) |
| Microscopy | | | | |
| Time (h) | 6.0 | 8.0 | 12.0 | |
| Cost (€) | 113.16 | 112.16 | 591.12 | 816.44 |
| RDT | | | | |
| Time (h) | 2.0 | 2.0 | 3.0 | |
| Cost (€) | 37.72 | 28.04 | 147.78 | 213.54 |
| LAMP | | | | |
| Time (h) | 3.0 | 3.0 | 4.0 | |
| Cost (€) | 56.58 | 42.06 | 197.04 | 295.68 |
| XN-31 | | | | |
| Time (h) | 4.0 | 4.0 | 4.0 | |
| Cost (€) | 75.44 | 56.08 | 197.04 | 328.56 |
| Operational execution of one blood test | | | | Costs/test(€) |
| Microscopy | | | | |
| Time (h) | 0.25 | 0.3 | 0.1 | |
| Cost (€) | 4.71 | 4.21 | 4.93 | 13.85 |
| RDT | | | | |
| Time (h) | 0.1 | 0.15 | 0.05 | |
| Cost (€) | 1.89 | 2.10 | 2.46 | 6.45 |
| LAMP | | | | |
| Time (h) | 0.3 | 0.1 | 0.05 | |
| Cost (€) | 5.658 | 1.402 | 2.463 | 9.523 |
| XN-31 | | | | |
| Time (h) | 0.05 | 0.05 | 0.05 | |
| Cost (€) | 0.94 | 0.70 | 2.46 | 4.10 |

RDT: rapid diagnosis test; LAMP: loop-mediated isothermal amplification.

TABLE III

Direct and indirect costs of methods for diagnosing malaria in France (2023)

|  | Microscopy | RDT | LAMP | XN-31 |
| --- | --- | --- | --- | --- |
| Direct costs (€)/test | 2.13 | 4.06 | 45.00 | 24.36 |
| Indirects costs/test | | | | |
| Training & QC cost/test | 0.82 | 0.21 | 0.30 | 0.33 |
| Execution test cost | 13.85 | 6.45 | 9.52 | 4.11 |
| Total indirect cost/test | 14.66 | 6.67 | 9.82 | 4.44 |
| Total cost (€)/test | 16.79 | 10.73 | 54.82 | 28.80 |

RDT: rapid diagnosis test; LAMP: loop-mediated isothermal amplification.

nance ratio of 1000 tests per year. Indirect costs were the addition of working costs based on the time needed for training, quality proficiency and operational execution needed to perform and validate each test, as recommended (24). The costs of repetitive testing during the post-treatment follow-up with microscopy were included in the analysis. Three tests were the minimum needed for the follow-up of an uncomplicated case (Days 3-7-28), and five tests were considered to be the minimum for the follow-up of a severe malaria case (Days 1-2-3-7-28). Follow-up cannot be conducted with RDT based on HRP2 detection or LAMP since these tests may be positive weeks after the resolution of the malaria case. Microscopy and XN-31 are the only methods able to provide information on the positive/negative status of the sample and the level of parasitaemia. In this study, we considered that follow-up should be performed with microscopy to reduce the global cost because post-treatment follow-up does not require a short time-to-result. Thus, the costs of three microscopy tests, noted "short follow-up", or five tests, noted "long follow-up", were added to each non-severe and severe malaria patient, respectively.

## RESULTS

*Study population* - The cohort of patients included on site during the year 2023 is described in Table IV. This cohort did not show significant differences from that in previous years. Most of the patients were people who originated from endemic areas and returned to France after visiting friends and relatives. The total number of samples tested for malaria diagnosis during the study period was 1199, corresponding to 809 samples collected at admission from patients suspected of having malaria and 390 samples tested during the follow-up of positive patients. During the year 2023, 192 malaria cases were confirmed (positivity rate: 23.7%). The final study population size for the cost-effectiveness study was the mean (1004) between tests at admission (809) and the total number of tests done (1199). Thus, the sample size for the cost-effectiveness study was approximately 1000/year.

At the French national level, the number of severe cases reported in 2023 by public health authorities in France (Malaria Reference Centre for Malaria) was 439 (44 children ≤ 15 years and 395 adults > 15 years). During the same period, 34 patients with severe malaria

TABLE IV

The epidemiological parameters for the analytical model are based on the cohort of patients included in 2023 at Lyon University Hospital (HCL)

| Parameters | Base case value (2023) |
| --- | --- |
| Total population | 809 |
| Mean age | 36.5 (15.32 - 57.68) |
| Median age | 31 (1 - 81) |
| Malaria cases | 192 (23.7%) |
| Non-malaria cases | 617 (76.2%) |
| *Falciparum* malaria | 155 (80.7%) |
| Non-*falciparum* malaria | 37 (19.3%) |
| Severe malaria cases | 8 (4.16%) |
| Non-severe malaria cases | 184 (95.8%) |
| Total samples including follow-up | 1199 (390 follow-up) |
| Positive samples | 382 (31.8%) |
| Negative samples | 817 (68.2%) |

(7.7% of the national incidence) were admitted to Lyon Hospital. Unfortunately, two of these patients died. The number of severe cases may be underestimated since isolated high parasitaemia cases are not systematically registered as severe in the absence of severe symptoms. No therapeutic failure was observed for either severe or non-severe patients during the study period.

*Cost-effectiveness analysis* - We developed a decision tree to compare the intervention to the three comparator scenarios. The incremental cost-effectiveness ratio (ICER) was used to compare the differences between the costs and health outcomes of the interventions and controls. We calculated incremental cost-effectiveness ratios as total incremental costs divided by total QALYs for the intervention compared to the control scenarios. The complete decision tree was built according to the methods section. The comparators were microscopy, RDT + microscopy and LAMP + microscopy. Invalid tests from the LAMP method were observed in our previous experience for 3% of the samples (13) and led to

retesting. Unknown species identification and abnormal scattergrams were observed with XN-31 for 5% of the tests (20) and required microscopy. The decision tree for the microscopy and intervention scenarios are presented (Figs 1-2). A cost-effectiveness plane was obtained for all the scenarios (Fig. 3). The cost-effectiveness results demonstrated that the intervention (XN-31 + microscopy) was dominant (most effective and least costly) to the comparators, RDT+microscopy and LAMP+microscopy. The intervention also stochastically dominates (first order) microscopy, while the basic cost of one XN-31 test is greater than that of microscopy. Indeed, the better performance of the XN-31 in terms of sensitivity and specificity and the reduced time needed for training were the basis for its cost-effectiveness.

*Sensitivity analysis* - Different parameters susceptible to heterogeneity in the cost-effectiveness analysis were tested with the same protocol. The dominance limit for the intervention was tested against the same comparators. The full cost of the intervention, initially calculated at approximately 30.00€ per test in France, was progressively increased to detect the threshold of cost-effectiveness, and the same model was run for each new value with the same comparators. The XN-31+microscopy scenario remains dominant with the cost increasing to 50.00€ per test.

Since LAMP test equipment and reagent costs can vary significantly, we also considered the variation in the total cost of LAMP tests based on the maximum discount obtained from the provider compared to catalogue prices. We used the lowest price obtained for equipment, and we ran the analysis model with a direct cost of LAMP of 30 € instead of 54.82 € per test. Using the same total cost for XN-31 and LAMP (approx. 30 €), the XN-31+M scenario was still dominant over the LAMP+M scenario.

## DISCUSSION

There are different highly efficient methods that can be used to diagnose cases of imported malaria in non-endemic areas. However, most of the national guidelines do not clarify the cost-effectiveness of these methods at the country level. The decision to select a scenario involving two complementary methods (screening and confirmation) rather than other scenarios is often made based on local expertise and on the direct costs proposed by local providers. The intuitive perception of diagnostic costs may be based on an incomplete consideration of the parameters that should be included in the total budget. However, the workload and work cost are two major parameters that are often overlooked in the decision-making process. While the workload involved in training and qualifying people who will perform these methods can be standardised to some extent, the work cost varies considerably, depending on the national regulations and qualifications required for individuals responsible for the biological diagnosis. This study is based on data from France only and cannot be directly applied to other countries or healthcare systems. Nevertheless, this study provides a global approach to determining the cost-effectiveness of the most common malaria diagnostic methods. This enables us to compare these methods and will assist final decision makers in selecting the most suitable option depending on local constraints.

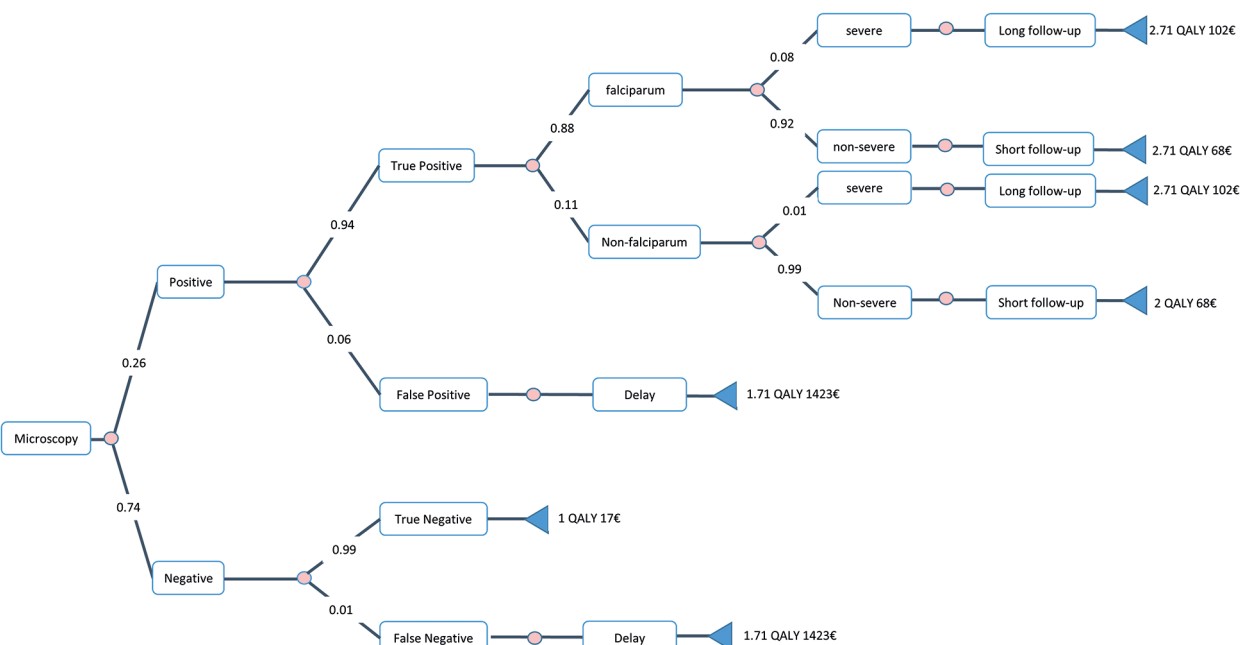

Fig. 1: decision tree generated from SpiceLogic software for microscopy scenario. The decision node on the left leads to a chance node with two possible outcomes: positive or negative. At each step, the result is either a true positive or a false negative. In the case of *Plasmodium falciparum*, the outcome can be either severe or non-severe, and the follow-up period should be either long or short. The frequency of each event is indicated on the respective branch.

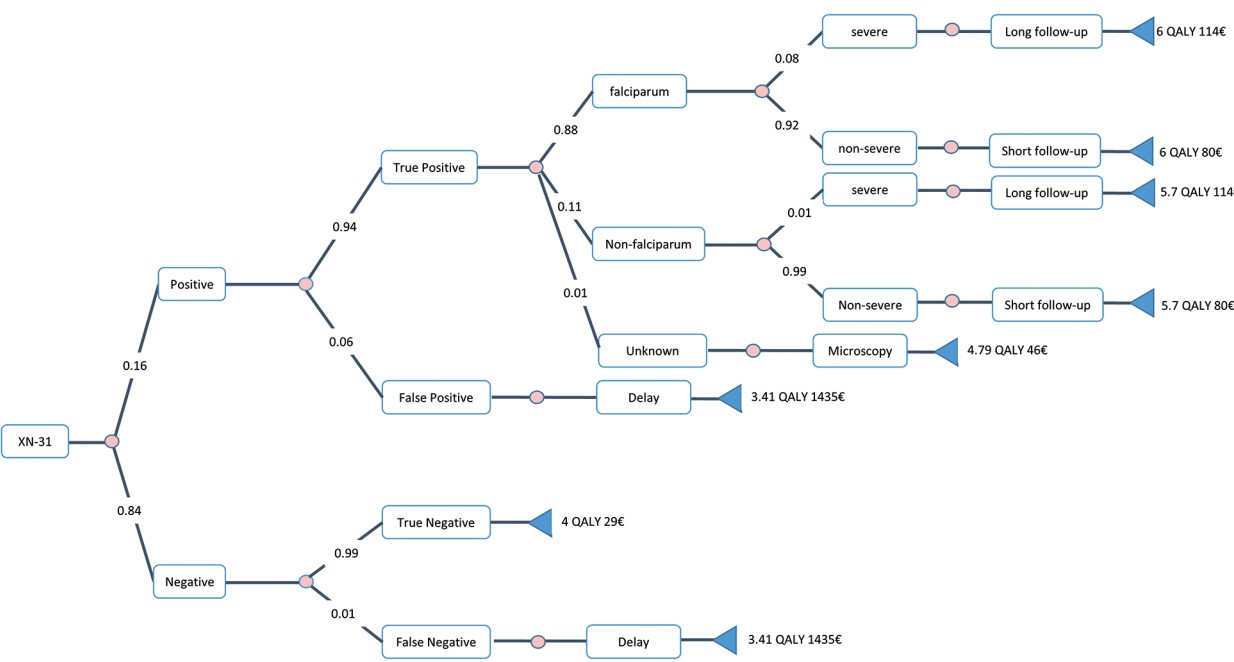

Fig. 2: decision tree generated from SpiceLogic software for XN-31 and microscopy scenario. The decision node on the left leads to a chance node with two possible outcomes: positive or negative (abnormal scattergrams are not represented here to reduce the complexity of the figure). At each step, the result is either a true positive or a false negative. In the case of *Plasmodium falciparum*, the outcome can be either severe or non-severe, and the follow-up period should be either long or short. The frequency of each event is indicated on the respective branch.

We observed a very small difference in the total RDT costs compared to microscopy. However, RDT-microscopy combination dominates LAMP-microscopy combination in a second order stochastic sense. This effect was due to the high direct cost of LAMP, which is greatly affected by the commercial value of reagents. In this study, we used costs identified from quotes provided by manufacturers. For the LAMP test in particular, there was a significant difference between the public sale costs and the potential cost reduction obtained from the provider. Nevertheless, the analytical model was run with a reduced LAMP price, and this heterogeneity did not alter the dominance of the scenario involving LAMP-microscopy combination.

It should be noted that while at least three or five tests were expected for non-severe and severe malaria patients, respectively, the real number of tests during the patient follow-up in 2023 in Lyon was far below [390 instead of 644 (non-severe: 158*3) + (severe: 34*5)]. The reason for this discrepancy between guidelines and reality are unclear. It can be speculated that the rapid efficacy of standard treatments based on artemisinin derivatives led to a short parasite clearance time.

This cost-effectiveness analysis is essential to determine the actual cost of malaria diagnosis in a non-endemic country based on the various scenarios.[32] Considering both the workload of each method and their respective diagnostic accuracy, the combination of historical microscopy and the recent XN-31 appears to be the most cost-effective option.

We also conducted a discrete event simulation to model the potential impact of the intervention on the critical time period between suspecting malaria and re-

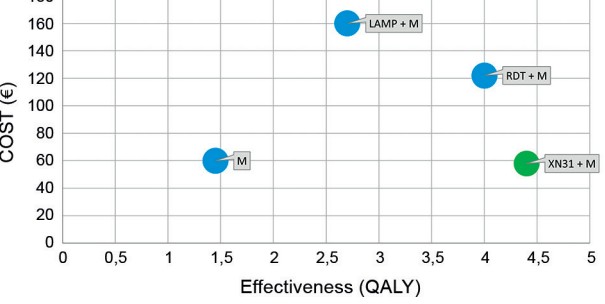

Fig. 3: cost effectiveness comparison of the different scenarios. (A) The cost-effectiveness ratio was calculated from the decision tree and graphically represented. The green plot represents the most cost-effective scenario. M: microscopy; RDT: rapid diagnosis test; LAMP: loop-mediated isothermal amplification.

ceiving anti-malarial treatment. It is hypothesised that a significant reduction in the time taken to receive specific antimalarial treatment is associated with a decreased risk of developing severe malaria.[33] A recent study conducted in France found that the median time from symptom onset to hospital arrival was three days [interquartile range (IQR) = 2-5].[34] Another study in Canada, showed that reducing the time to first-dose treatment from eight to five and a half after admission to hospital was associated with a reduction in the number of ICU passages, which may be multifactorial.[35] Thus, there is evidence that reducing the time taken to make a diagnosis can improve outcomes. In France, the delay between blood sampling and the laboratory receiving the sample is legally restricted to 2 h. This is followed by the time

needed to perform the analysis and provide the result to the clinician, which should take no longer than 2 more hours. In our experience, 98% of results are provided in under 2 h, which is the time required to perform RDT and LAMP tests to distinguish between negative and positive samples. Species identification and parasitaemia determination require an additional hour. Overall, the time between blood sampling and transmission of the complete result is a maximum of 4 h for a negative result, and 5 h for species identification and parasitaemia count of a positive sample. Using the XN-31 will result in a time saving of approximately 2 h, as there is no need for sample preparation, such as staining blood smears or extracting DNA. The test takes one minute to provide positive/negative, *falciparum*/non-*falciparum* and parasitaemia results. A 2-hour reduction in time to diagnosis may be associated with a reduction in the risk of severe or lethal malaria. However, given that the rate of severe cases in the included population is low, the potential reduction in severity should be marginal, with no significant impact on the overall cost of malaria diagnosis. While the reduction in time to diagnosis cannot be measured as a net monetary benefit, it is a compelling argument in favour of the intervention scenario for clinicians, decision-makers and stakeholders.

This study has some significant limitations. Firstly, the information collected here is locally sourced, which makes it difficult to compare with other countries. While the study's value is strictly limited to France, it could inform similar approaches in other settings. The second limitation is that the workload associated with each tested method was estimated based on interviews. A more robust approach should have involving measuring the exact time needed to completely perform the tests, the training and quality assurance in full, but this would have depended on local expertise. A third limitation is the systematic requirement for microscopy to identify species for all methods. This was included in the decision tree modelling. It should be noted that the values obtained were derived from complex scenarios rather than single methods. The fourth limitation is the absence of a comparison with PCR, as the focus was on the cost of methods that provide emergency results for patients rather on identifying the most effective method of detecting and identifying *Plasmodium*. Finally, the financial support from the provider of the innovation was not associated with any interference in the data collection, analysis or reporting.

### ACKNOWLEDGEMENTS

To the technicians, junior and senior doctors involved in malaria diagnosis for their participation in the workload evaluation for training, quality proficiency and test execution.

### AUTHORS' CONTRIBUTION

SP - conception of the study, design, data, analysis, writting of first draft, writting of final draft; ALB - conception of the study, design, data, analysis, reviewing of first draft, reviewing of final draft. Sysmex had no impact on study design, results and discussion. The raw data collected during the study and presented in this paper were not disclosed to Sysmex until after the authors had conducted their analysis and reached a conclusion regarding the cost-effectiveness of the different scenarios tested. Sysmex has no access to Spicelogic data sheets and could not influence the software analysis.

### DATA AVAILABILITY

The contents underlying the research text are included in the manuscript.

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

# OPEN PEER REVIEW

Memórias do IOC thanks the anonymous reviewers for their contribution to the peer review of this work.

**FIRST REVIEW ROUND**

REVIEWERS' COMMENTS

## REVIEWER #1

Reviewer comments: Manuscript Title: Comparison of the cost-effectiveness of the current standard of care for diagnosing malaria with that of Sysmex XN-31 in a nonendemic country"

Authors: Stephane Picot & Anne-Lise Bienvenu

This manuscript aims to provide the cost-effectiveness analysis of malaria diagnostic strategies in France, a non-endemic country, comparing conventional methods (microscopy, RDT, LAMP) with the Sysmex XN-31 hematology analyzer, concluding that the XN-31, in combination with microscopy, is the most cost-effective strategy. While the topic is highly relevant and timely, the manuscript presents significant methodological limitations and potential biases that compromise the strength and generalizability of its conclusions.

Major Issues:

1. The authors state that the study is supported by Sysmex Corporation, manufacturer of the evaluated diagnostic tool. Despite a statement of non-involvement, the strong endorsement of XN-31 raises concerns of potential bias. A clearer, more detailed conflict of interest declaration is necessary, and independent validation would enhance credibility and mitigate bias.

2. All data are originate from a single institution, Lyon University Hospital, using internal estimates for sensitivity, specificity, operational time, and training requirements derived from previous studies by the same group. These localized parameters limit the generalizability of the findings, especially for decision-makers in other healthcare systems. This information should be highlighted in the manuscript.

3. Estimates for training time, quality proficiency, and operational labor were derived from internal interviews without any external comparison. This may exaggerate the cost burden of microscopy and favor automated solutions.

4. Although the authors state that most of the laboratories in nonendemic countries use a combination of microscopy, RDTs and LAMP or PCR according to French National guidelines, there is no comparison with PCR, a widely recognized reference method for malaria diagnosis due to its high sensitivity and specificity, and considered the gold standard in many reference laboratories, especially for detecting low-level parasitemia and mixed infections. So, it would be highly valuable to include a comparison between XN-31 and PCR in terms of performance and cost-effectiveness, ideally using available data from the same setting or from previous studies. If PCR data were not available, the authors can justify the exclusion and to discuss its known advantages and limitations compared to XN-31, using the existing literature. Including this comparison would strengthen the manuscript's relevance and ensure that all major diagnostic modalities are adequately addressed.

5. While the manuscript highlights the performance of the XN-31 its inability to reliably identify non-falciparum species is briefly acknowledged but insufficiently modeled. Since species identification is essential for guiding treatment (e.g., anti-relapse therapy for P. vivax), this represents a critical point, and this limitation should have been incorporated into the cost-effectiveness model.

Minor Issues:

• The manuscript contains grammatical and typographical errors, and a full language review is necessary.

• The figures, especially the decision trees, are of poor quality, making it difficult to visualize the data and interpret the results.

## REVIEWER #2

Reviewer comments: The study aimed to perform a detailed cost-effectiveness analysis and compare various scenarios in France. Therefore, please focus on this and avoid repetition throughout the manuscript.

Why did you choose to investigate loop-mediated isothermal amplification (LAMP) instead of polymerase chain reaction (PCR)? Please add an explanation for this choice and consider comparing results with those of PCR.

The number of Figures (7) and Tables (6) seems excessive. Would it be possible to reduce them?

How does the sensitivity of the RDT, LAMP, and XN-31 compare to parasitemia levels determined by microscopic examination?

Could you please add a sentence about all the limitations of the study in the Discussion section?

LINES 45-47: Total cost (direct + indirect) of biological diagnosis (controls and intervention):

….." Follow-up cannot be conducted with RDT or LAMP since these tests may be positive weeks after the resolution of the malaria case."

This does not apply to RDT targeting DLDH antigens, as this protein is only expressed by live parasites. Please rewrite this sentence.

LINES 50-55 Page 7:

The authors concluded that" it is recommended to address the need to prevent the recurrence of Plasmodium vivax via specific treatment with primaquine after the resolution of active malaria cases". It means that you should use another test associated with XN-31. Have you evaluated this need in terms of cost-effectiveness? Please clarify this point on the MS.

## AUTHORS' RESPONSE TO THE REVIEWERS

We would like to express our gratitude to the reviewers for their time in evaluating our work and providing feedback to enhance it. The main changes required by the reviewers are highlighted in yellow in the manuscript. The english has been improved.

Reviewer 1:

1: The authors state that the study is supported by Sysmex Corporation, manufacturer of the evaluateddiagnostic tool. Despite a statement of non-involvement, the strong endorsement of XN-31 raises concerns of potential bias. A clearer, more detailed conflict of interest declaration is necessary, and independent validationwould enhance credibility and mitigate bias.

Answer: We agree with the reviewer that support from the company could raise concerns about bias, and we took this into account when preparing the manuscript. The budget required to conduct this study could not be obtained from an academic call, since the objective falls outside the scope of almost all calls. However, many studies of new diagnostic methods, drugs or interventions receive financial or technical support from their providers. This needs to be clearly indicated and we did mention the support from the company in the draft. We are unsure what a 'clearer, more detailed conflict of interest declaration' would entail. In response to this query, we have addressed the issue in the text and at the end of the manuscript. We have also deleted any sentences that could be interpreted as a strong endorsement of the method. We declare that our results are based on an objective assessment of the information gathered and the results obtained.

2: All data are originate from a single institution, Lyon University Hospital, using internal estimates for sensitivity, specificity, operational time, and training requirements derived from previous studies by the same group. These localized parameters limit the generalizability of the findings, especially for decision-makers in other healthcare systems. This information should be highlighted in the manuscript.

Answer: The limitation of this study to french healthcare systems was already mentioned throughout the manuscript:

• in the abstract
• in the text (page 7 (data source from the French Technical agency ; cost paid by the French government) , page 8 (price paid by the Lyon Hospital , table 2: methods in France
• In the discussion: Page 22, ligne 4 : "The present study is based on data available in France only and should not be directly translated to other countries or other health care systems"

The abstract, discussion and limitation section of the paper all reiterate this point.Changes have been highlighted in the text.

3. Estimates for training time, quality proficiency, and operational labor were derived from internal interviews without any external comparation. This may exaggerate the cost burden of microscopy and favor automated solutions.

Answer: We agree that this parameter is mainly based on professional experience in the lab. However, it is not surprising that the microscopy workload, including training and quality assurance, is higher than that of automated methods. Indeed, the cost of human resources for qualified microscopy in non-endemic areas is one of the drivers behind for the development of automated methods.

4. … it would be highly valuable to include a comparison between XN-31and PCR in terms of performance and cost-effectiveness, ideally using available data from the same setting or from previous studies. If PCR data were not available, the authors can justify the exclusion and to discuss its known advantages and limitations compared to XN-31…

Answer: The aim of this study was to compare the methods used for the initial diagnosis of malaria, which requires a rapid result. In France, where the study was designed and conducted, the legally acceptable timeframe for providing results to the prescriber after receiving a sample in the laboratory is two hours. The diagnostic methods included in this study met this mandatory requirement. PCR is not available 24/7 and thus cannot be used for this purpose. In our experience, PCR is only used to confirm very low parasitaemia or species identification after the initial diagnosis has been obtained. This statement has been added to the background section on page 2 of the manuscript.

5. While the manuscript highlights the performance of the XN-31 its inability to reliably identify non-falciparum species is briefly acknowledged but insufficiently modeled. Since species identification is essential for guiding treatment (e.g., anti-relapse therapy for P. vivax), this represents a critical point, and this limitation should have been incorporated into the cost-effectiveness model.

Answer: As described in the manuscript, the requirement for species identification following a positive test result was incorporated into the model. Indeed, the additional cost of microscopy was systematically summed with the cost of XN-31. This study did not compared the cost-effectiveness of the XN method to other methods, but rather compared different scenarios involving an automated method plus microscopy. This point has been clarified throughout the manuscript and in the discussion section. The need for species identification was clearly acknowledged in the background section on page 3 and in the method section on page 4.

Reviewer 2:

1. Why did you choose to investigate LAMP instead of PCR?

Answer: Please see answer 4 to reviewer 1

2. The number of figures (7) and tables (6) seems excessive. Would it be possible to reduce them?

Answer : We agree with this comment, thank you. The number of tables and figures has been significantly reduced to 4 tables and 3 figures. The quality of the figures has been improved to reach 300 dpi in TIFF format.

3. How does the sensitivity of the RDT, LAMP and XN31 compare to parasitemia levels determined by microscopic examination?

Answer: The XN31 correlate well with the parasitemia obtained with microscopy, as demonstrated in the previous paper: Diagnostic accuracy of fluorescence flow-cytometry technology using Sysmex XN-31 for imported malaria in a non-endemic setting. Picot S, Perpoint T, Chidiac C, Sigal A, Javouhey E, Gillet Y, Jacquin L, Douplat M, Tazarourte K, Argaud L, Wallon M, Miossec C, Bonnot G, Bienvenu AL. Parasite. 2022;29:31. doi: 10.1051/parasite/2022031. Epub 2022 May 31. This reference has been included in the text.

4. Could you please add a sentence about all the limitations of the study in the Discussion section?

Answer: Yes, this was clearly missing, thank you. A paragraph about all the limitations was added at the end of the discussion section.

LINES 45-47: Total cost (direct + indirect) of biological diagnosis (controls and intervention): …..” Follow-up cannot be conducted with RDT or LAMP since these tests may be positive weeks after the resolution of the malaria case.” This does not apply to RDT targeting DLDH antigens, as this protein is only expressed by live parasites. Please rewrite this sentence.

Answer : Thank you for your input. The precision was included in the text on page 13.

LINES 50-55 Page 7:

The authors concluded that” it is recommended to address the need to prevent the recurrence of Plasmodium vivax via specific treatment with primaquine after the resolution of active malaria cases”. It means that you should use another test associated with XN-31. Have you evaluated this need in terms of cost-effectiveness?

Please clarify this point on the MS

Answer: Please see answer 5 to reviewer 1. Thank you.

## SECOND REVIEW ROUND

### REVIEWERS' COMMENTS

**REVIEWER #1**

Reviewer comments: Accept

**REVIEWER #2**

Reviewer comments: After the revision by the authors, the paper can now be accepted for publication.

