## [Reviewer Report · FIRST REVIEW ROUND - REVIEWERS COMMENTS]

## REVIEWER #1

Reviewer comments: Manuscript Title: Comparison of the cost-effectiveness of the current standard of care for diagnosing malaria with that of Sysmex XN-31 in a nonendemic country”

Authors: Stephane Picot e Anne-Lise Bienvenu

This manuscript aims to provide the cost-effectiveness analysis of malaria diagnostic strategies in France, a non-endemic country, comparing conventional methods (microscopy, RDT, LAMP) with the Sysmex XN-31 hematology analyzer, concluding that the XN-31, in combination with microscopy, is the most cost-effectiveness strategy. While the topic is highly relevant and timely, the manuscript presents significant methodological limitations and potential biases that compromise the strength and generalizability of its conclusions.

Major Issues:

1. The authors state that the study is supported by Sysmex Corporation, manufacturer of the evaluated diagnostic tool. Despite a statement of non-involvement, the strong endorsement of XN-31 raises concerns of potential bias. A clearer, more detailed conflict of interest declaration is necessary, and independent validation would enhance credibility and mitigate bias.

2. All data are originate from a single institution, Lyon University Hospital, using internal estimates for sensitivity, specificity, operational time, and training requirements derived from previous studies by the same group. These localized parameters limit the generalizability of the findings, especially for decision-makers in other healthcare systems. This information should be highlighted in the manuscript.

3. Estimates for training time, quality proficiency, and operational labor were derived from internal interviews without any external comparation. This may exaggerate the cost burden of microscopy and favor automated solutions.

4. Although the authors state that most of the laboratories in nonendemic countries use a combination of microscopy, RDTs and LAMP or PCR according to French National guidelines, there is no comparison with PCR, a widely recognized reference method for malaria diagnosis due to its high sensitivity and specificity, and considered the gold standard in many reference laboratories, especially for detecting low-level parasitemia and mixed infections. So, it would be highly valuable to include a comparison between XN-31 and PCR in terms of performance and cost-effectiveness, ideally using available data from the same setting or from previous studies. If PCR data were not available, the authors can justify the exclusion and to discuss its known advantages and limitations compared to XN-31, using the existing literature. Including this comparison would strengthen the manuscript’s relevance and ensure that all major diagnostic modalities are adequately addressed.

5. While the manuscript highlights the performance of the XN-31 its inability to reliably identify non-falciparum species is briefly acknowledged but insufficiently modeled. Since species identification is essential for guiding treatment (e.g., anti-relapse therapy for P. vivax), this represents a critical point, and this limitation should have been incorporated into the cost-effectiveness model.

Minor Issues:

• The manuscript contains grammatical and typographical errors, and a full language review is necessary.

• The figures, especially the decision trees, are of poor quality, making it difficult to visualize the data and interpret the results.

## REVIEWER #2

Reviewer comments: The study aimed to perform a detailed cost-effectiveness analysis and compare various scenarios in France. Therefore, please focus on this and avoid repetition throughout the manuscript.

Why did you choose to investigate loop-mediated isothermal amplification (LAMP) instead of polymerase chain reaction (PCR)? Please add an explanation for this choice and consider comparing results with those of PCR.

The number of Figures (7) and Tables (6) seems excessive. Would it be possible to reduce them?

How does the sensitivity of the RDT, LAMP, and XN-31 compare to parasitemia levels determined by microscopic examination?

Could you please add a sentence about all the limitations of the study in the Discussion section?

LINES 45-47: Total cost (direct + indirect) of biological diagnosis (controls and intervention):

…..” Follow-up cannot be conducted with RDT or LAMP since these tests may be positive weeks after the resolution of the malaria case.”

This does not apply to RDT targeting DLDH antigens, as this protein is only expressed by live parasites. Please rewrite this sentence.

LINES 50-55 Page 7:

The authors concluded that” it is recommended to address the need to prevent the recurrence of Plasmodium vivax via specific treatment with primaquine after the resolution of active malaria cases”. It means that you should use another test associated with XN-31. Have you evaluated this need in terms of cost-effectiveness? Please clarify this point on the MS.

---

## [Author Response · AUTHORS RESPONSE TO REVIEWERS]

## AUTHORS’ RESPONSE TO THE REVIEWERS

We would like to express our gratitude to the reviewers for their time in evaluating our work and providing feedback to enhance it. The main changes required by the reviewers are highlighted in yellow in the manuscript. The english has been improved.

## Reviewer 1

1: The authors state that the study is supported by Sysmex Corporation, manufacturer of the evaluateddiagnostic tool. Despite a statement of non-involvement, the strong endorsement of XN-31 raises concerns of potential bias. A clearer, more detailed conflict of interest declaration is necessary, and independent validationwould enhance credibility and mitigate bias.

Answer: We agree with the reviewer that support from the company could raise concerns about bias, and we took this into account when preparing the manuscript. The budget required to conduct this study could not be obtained from an academic call, since the objective falls outside the scope of almost all calls. However, many studies of new diagnostic methods, drugs or interventions receive financial or technical support from their providers. This needs to be clearly indicated and we did mention the support from the company in the draft. We are unsure what a ‘clearer, more detailed conflict of interest declaration’ would entail. In response to this query, we have addressed the issue in the text and at the end of the manuscript. We have also deleted any sentences that could be interpreted as a strong endorsement of the method. We declare that our results are based on an objective assessment of the information gathered and the results obtained.

2: All data are originate from a single institution, Lyon University Hospital, using internal estimates for sensitivity, specificity, operational time, and training requirements derived from previous studies by the same group. These localized parameters limit the generalizability of the findings, especially for decision-makers in other healthcare systems. This information should be highlighted in the manuscript.

Answer: The limitation of this study to french healthcare systems was already mentioned throughout the manuscript:

• in the abstract

• in the text (page 7 (data source from the French Technical agency ; cost paid by the French government) , page 8 (price paid by the Lyon Hospital , table 2: methods in France

• In the discussion: Page 22, ligne 4 : “The present study is based on data available in France only and should not be directly translated to other countries or other health care systems”

The abstract, discussion and limitation section of the paper all reiterate this point.Changes have been highlighted in the text.

3. Estimates for training time, quality proficiency, and operational labor were derived from internal interviews without any external comparation. This may exaggerate the cost burden of microscopy and favor automated solutions.

Answer: We agree that this parameter is mainly based on professional experience in the lab. However, it is not surprising that the microscopy workload, including training and quality assurance, is higher than that of automated methods. Indeed, the cost of human resources for qualified microscopy in non-endemic areas is one of the drivers behind for the development of automated methods.

4. … it would be highly valuable to include a comparison between XN-31and PCR in terms of performance and cost-effectiveness, ideally using available data from the same setting or from previous studies. If PCR data were not available, the authors can justify the exclusion and to discuss its known advantages and limitations compared to XN-31…

Answer: The aim of this study was to compare the methods used for the initial diagnosis of malaria, which requires a rapid result. In France, where the study was designed and conducted, the legally acceptable timeframe for providing results to the prescriber after receiving a sample in the laboratory is two hours. The diagnostic methods included in this study met this mandatory requirement. PCR is not available 24/7 and thus cannot be used for this purpose. In our experience, PCR is only used to confirm very low parasitaemia or species identification after the initial diagnosis has been obtained. This statement has been added to the background section on page 2 of the manuscript.

5. While the manuscript highlights the performance of the XN-31 its inability to reliably identify non-falciparum species is briefly acknowledged but insufficiently modeled. Since species identification is essential for guiding treatment (e.g., anti-relapse therapy for P. vivax), this represents a critical point, and this limitation should have been incorporated into the cost-effectiveness model.

Answer: As described in the manuscript, the requirement for species identification following a positive test result was incorporated into the model. Indeed, the additional cost of microscopy was systematically summed with the cost of XN-31. This study did not compared the cost-effectiveness of the XN method to other methods, but rather compared different scenarios involving an automated method plus microscopy. This point has been clarified throughout the manuscript and in the discussion section. The need for species identification was clearly acknowledged in the background section on page 3 and in the method section on page 4.

## Reviewer 2

1. Why did you choose to investigate LAMP instead of PCR?

Answer: Please see answer 4 to reviewer 1

2. The number of figures (7) and tables (6) seems excessive. Would it be possible to reduce them?

Answer : We agree with this comment, thank you. The number of tables and figures has been significantly reduced to 4 tables and 3 figures. The quality of the figures has been improved to reach 300 dpi in TIFF format.

3. How does the sensitivity of the RDT, LAMP and XN31 compare to parasitemia levels determined by microscopic examination?

Answer: The XN31 correlate well with the parasitemia obtained with microscopy, as demonstrated in the previous paper: Diagnostic accuracy of fluorescence flow-cytometry technology using Sysmex XN-31 for imported malaria in a non-endemic setting. Picot S, Perpoint T, Chidiac C, Sigal A, Javouhey E, Gillet Y, Jacquin L, Douplat M, Tazarourte K, Argaud L, Wallon M, Miossec C, Bonnot G, Bienvenu AL. Parasite. 2022;29:31. doi: 10.1051/parasite/2022031. Epub 2022 May 31. This reference has been included in the text.

4. Could you please add a sentence about all the limitations of the study in the Discussion section?

Answer: Yes, this was clearly missing, thank you. A paragraph about all the limitations was added at the end of the discussion section.

LINES 45-47: Total cost (direct + indirect) of biological diagnosis (controls and intervention): …..” Follow-up cannot be conducted with RDT or LAMP since these tests may be positive weeks after the resolution of the malaria case.” This does not apply to RDT targeting DLDH antigens, as this protein is only expressed by live parasites. Please rewrite this sentence.

Answer : Thank you for your input. The precision was included in the text on page 13.

LINES 50-55 Page 7:

The authors concluded that” it is recommended to address the need to prevent the recurrence of Plasmodium vivax via specific treatment with primaquine after the resolution of active malaria cases”. It means that you should use another test associated with XN-31. Have you evaluated this need in terms of cost-effectiveness?

Please clarify this point on the MS

Answer: Please see answer 5 to reviewer 1. Thank you.

---

## [Reviewer Report · REVIEWERS COMMENTS]

## REVIEWER #1

Reviewer comments: Accept

## REVIEWER #2

Reviewer comments: After the revision by the authors, the paper can now be accepted for publication.